# Losartan Prevents Hepatic Steatosis and Macrophage Polarization by Inhibiting HIF-1α in a Murine Model of NAFLD

**DOI:** 10.3390/ijms22157841

**Published:** 2021-07-22

**Authors:** Cheng-Hui Wang, Hsuan-Miao Liu, Zi-Yu Chang, Tse-Hung Huang, Tzung-Yan Lee

**Affiliations:** 1Graduate Institute of Clinical Medical Sciences, College of Medicine, Chang Gung University, Taoyuan City 333, Taiwan; risingsun4518@gmail.com; 2Graduate Institute of Traditional Chinese Medicine, School of Chinese Medicine, College of Medicine, Chang Gung University, Taoyuan City 333, Taiwan; miaowhale@gmail.com; 3Department of Traditional Chinese Medicine, Chang Gung Memorial Hospital, Keelung City 204, Taiwan; changzhi887@gmail.com (Z.-Y.C.); huangtsehung@gmail.com (T.-H.H.); 4Institute of Traditional Medicine, School of Medicine, National Yang-Ming University, Taipei 112, Taiwan

**Keywords:** losartan, non-alcoholic fatty liver disease, HIF-1α, lipid droplet, macrophage polarization

## Abstract

Hypoxia and hepatosteatosis microenvironments are fundamental traits of nonalcoholic fatty liver disease (NAFLD). Hypoxia-inducible factor-1α (HIF-1α) is a transcription factor that controls the cellular response to hypoxia and is activated in hepatocytes of patients with NAFLD, whereas the route and regulation of lipid droplets (LDs) and macrophage polarization related to systemic inflammation in NAFLD is unknown. Losartan is an angiotensin II receptor antagonist, that approved portal hypertension and related HIF-1α pathways in hepatic injury models. Here, we show that losartan in a murine model of NAFLD significantly decreased hepatic de novo lipogenesis (DNL) as well as suppressed lipid droplets (LDs), LD-associated proteins, perilipins (PLINs), and cell-death-inducing DNA-fragmentation-factor (DFF45)-like effector (CIDE) family in liver and epididymal white adipose tissues (EWAT) of *ob*/*ob* mice. Obesity-mediated macrophage M1 activation was also required for HIF-1α expression in the liver and EWAT of *ob*/*ob* mice. Administration of losartan significantly diminishes obesity-enhanced macrophage M1 activation and suppresses hepatosteatosis. Moreover, HIF-1α-mediated mitochondrial dysfunction was reversed in *ob*/*ob* mice treated with losartan. Together, the regulation of HIF-1α controls LDs protein expression and macrophage polarization, which highlights a potential target for losartan in NAFLD.

## 1. Introduction

Obesity-related steatosis and nonalcoholic fatty liver disease (NAFLD) are associated with hepatic steatosis or accumulation of fat, predominantly triglycerides, that promote mitochondrial dysfunction and M1 macrophage polarization [1,2]. Obesity triggers hypoxia in adipose tissue and the small intestine, which stabilizes and activates hypoxia-inducible factor-1α (HIF-1α), a transcription factor implicated in hypoxia and hepatic lipid accumulation, resulting in adverse metabolic effects, including insulin resistance and NAFLD [1,3,4]. Recent findings demonstrate that HIF-1α is implicated in lipid metabolism through LDs accumulation [5], an increase of fatty acid and lipid synthesis [5,6], and upregulation of fatty acid uptake [7].

Lipid droplets (LDs) play central roles in cellular and organismal energy homeostasis, in particular, and overall lipid metabolism in general, that are particularly important in tissues specialized for energy storage or lipid turnovers, such as adipose tissues, the liver, and the intestine [8,9]. On the surface and/or in the vicinity of droplets, there are several structural/functional proteins such as lipid droplet proteins, lipogenic enzymes, and lipases, as well as LD-associated proteins such as perilipin families (PLIN1-PLIN5) and cell death-inducing DNA fragmentation factor-alpha (DFFA)-like effector (CIDE) proteins. PLIN1 tended to be associated with larger LDs, while expression of PLIN2 was more correlated with the presence of ballooned hepatocytes and evidence of oxidation [10].

An interesting mechanism recently related to the pathogenesis of NAFLD is the association between mitochondrial dysfunction and hepatocyte sensitivity to hypoxic stress. A high-fat diet induces dysregulation of hepatic oxygen gradients and mitochondrial function in vivo by reduced complex IV activity [11]. Lipid droplets and mitochondria indeed display close physical associations [12], and direct channeling of fatty acids from their site of release (droplets) to the site of consumption (mitochondria) would minimize the risk of toxic effects elsewhere, such as disruption of cellular membranes or inappropriate nuclear signaling [13,14]. When fusion is prevented, mitochondria are fragmented, and efficient uptake of fatty acids and their metabolic breakdown occurs only in the mitochondria directly associated with lipid droplets. Unmetabolized fatty acids are re-exported, either into lipid droplets or into the extracellular space, resulting in lipid accumulation and obesity [13,14,15].

In obesity, elevated fatty acids are a potential trigger for macrophage activation [16], which play an important role in the modulation of inflammation through polarizing to classically activated macrophages (M1) or alternatively activated macrophages (M2) in certain tissue niches and upon environmental stimuli [17]. Thus, M1-type macrophages, secrete pro-inflammatory cytokines such as tumor necrosis factor (TNF) α and interleukin (IL) -6, are predominant adipose tissue macrophage (ATM) populations in obese adipose tissues [18]. Previous studies have demonstrated that HIF-1α is a metabolic regulator that plays an important role in immunologic responses and participates in the M1 polarization of macrophages [19], and Ouyang X et al. has shown that inhibiting HIF-1 α activation suppresses liver inflammation and cellular damage in steatohepatitis [20].

Losartan (Cozaar) is an angiotensin II receptor antagonist, that has been known to attenuate progression of nonalcoholic steatohepatitis in obese models [21,22,23]. Our early studies demonstrate that losartan decreased portal pressure and ameliorated hyperdynamic circulation on bile duct-ligated cirrhotic rats with portal hypertension [24] and involvement of the HIF-1α and Wnt/β-catenin pathways on fatty liver graft with ischemia/reperfusion injury [25]. In additional, Yang et al. demonstrated that losartan alleviated angiotensin II induced-lipid droplet (LD) accumulation and expression of the LD marker adipose differentiation-related protein (ADRP) in podocytes [26]. However, the effects of losartan on HIF-1α, LDs and mitochondrial function have not yet been completely understood. In this study, we demonstrate that losartan enhances hepatic mitochondrial biogenesis/function and M2 macrophage polarization by inhibiting HIF-1α and lipogenesis pathways. Further, losartan was significantly mitigated by LDs accumulation and M1 macrophage polarization in EWAT, suggesting potential therapeutic applications for the treatment of obesity-related diseases.

## 2. Results

### 2.1. Losartan Prevents NAFLD Development

We evaluated the effects of losartan on NAFLD in obese models and compared their effects to improved liver function after 30 days of drug treatment. *ob*/*ob* mice displayed a significantly increased body weight, and the ratio of liver/ body weight index was compared with normal mice (Figure 1A). Although no significant differences in body weight were observed between *ob*/*ob* mice and *ob*/*ob* mice treated losartan, losartan significantly reduced the ratio of liver/body weight in *ob*/*ob* mice (Figure 1A,B). No significant differences in plasma alanine aminotransferase (ALT) and aspartate transaminase (AST) were observed between *ob*/*ob* mice and *ob*/*ob* mice treated losartan (Figure 1C,D). Triglyceride (TG) and free fatty acid (FFA) levels were significantly increased in *ob*/*ob* mice as compared with normal mice and were significantly reduced in *ob*/*ob* mice administrated losartan (Figure 1E,F), suggesting that losartan has a potential effect of preventing NAFLD development.

### 2.2. Losartan Attenuates Liver Steatosis and Hepatic HIF-1α Activation

LDs are dynamic lipid storage organelles that are found in hepatocytes on *ob*/*ob* mice. Lack of leptin led to profound modifications in hepatocyte morphology and physiology (Figure 2). Interestingly, these changes were fat-depot specific. We assessed the abundance of lipid accumulation in hepatocytes on *ob*/*ob* mice, a lipophilic dye on the Oil Red O (Figure 2A). These data were further confirmed by HE and Oil Red O staining, highlighting a novel role for losartan in reducing LDs accumulation in the context of obesity (Figure 2A). As shown in Figure 1A,B, the HIF-1α protein level was significantly decreased by losartan treatment in *ob*/*ob* mice (Figure 2A,B). Hepatocytes of *ob*/*ob* mice were larger and contained large multilocular LDs (Figure 2A), and showed that LD-associated proteins (PLIN1, PLIN2, CIDEA and CIDEC) and lipofuscin were significantly increased in the liver (Figure 2A,B). Losartan significantly reduced LD-associated proteins and lipofuscin in *ob*/*ob* mice (Figure 2A,B). Moreover, lipolysis enzymes as known to regulate basal lipolysis and lipid droplet size. As shown in Figure 2C, losartan significantly reduced lipolysis enzyme mRNA expression, including adipose triglyceride lipase (ATGL), hormone-sensitive lipase (HSL), lipoprotein lipase (LPL) and acyl-CoA oxidase (ACO) in the liver of *ob*/*ob* mice (Figure 2C).

### 2.3. Losartan Reduces Hepatic Lipogenesis

We next investigated cellular mechanisms for the effects of losartan on hepatic lipid accumulation. *ob*/*ob* mice showed a large increase in LD levels, probably via induction hepatic lipogenesis through sterol regulatory element binding protein 1 (SREBP-1) activation and fatty acid uptake cluster of differentiation 36 (CD36) expression. Losartan markedly decreased SREBP-1 and CD36 protein levels (Figure 3A,B) as well as SREBP-1c, fatty acid synthase (FAS), stearoyl-CoA desaturase-1 (SCD-1), and fatty acid transporters (CD36 and fatty acid transport protein, FATP) mRNA expression in the liver of *ob*/*ob* mice (Figure 3C). These data indicate that losartan decreased de novo lipogenesis (DNL) and fatty acid (FA) uptake via HIF-1α and SREBP-1c-dependant pathways.

### 2.4. Losartan Improves Hepatic Mitochondrial Biogenesis and β-Oxidation

We aimed to explore losartan effects on the lipid relationship between hepatic mitochondrial biogenesis and β-oxidation in *ob*/*ob* mice. Hepatic mitochondria biogenesis markers, sirtuin-1 (SIRT1), peroxisome proliferator-activated receptor gamma coactivator 1 alpha (PGC1α), uncoupling protein 1 (UCP1), and UCP2 were significantly decreased in *ob*/*ob* mice (Figure 4A,B). Gene expressions of PGC1α, nuclear respiratory factor (NRF) 1, NRF2 and mitochondrial transcription factor A (TFAM) were also reduced in *ob*/*ob* mice (Figure 4C). During losartan treatment in *ob*/*ob* mice, mitochondria biogenesis gradually returned to those observed in normal mice (Figure 4A–C). Next, we observed high levels of PPARα, carnitine palmitoyltransferase (CPT)-1, CPT-2, long-chain acyl-CoA dehydrogenase (LCAD), and medium-chain acyl-CoA dehydrogenase (MCAD) mRNA expression in *ob*/*ob* mice, and we observed significantly reduced PPARα and β-oxidation markers in liver of *ob*/*ob* mice treatment losartan (Figure 4D).

### 2.5. Losartan Attenuates Hepatic Inflammatory Cytokine mRNA Expression and Relate Hepatic Macrophage Polarization

We next investigated cellular mechanisms for the effects of losartan on hepatic macrophage polarization. Losartan treatment decreased monocyte chemoattractant protein-1 (MCP-1), tumor necrosis factor α (TNFα), and interferon gamma (IFNγ) mRNA expression in *ob*/*ob* mice (Figure 5A). We performed immunohistochemical staining of liver-infiltrating cells. When compared with a normal liver, M1 macrophage (CD11b, CD11c, and CCR7) expression was significantly increased, and M2 macrophages (CD163, and CD206) were reduced in *ob*/*ob* mice (Figure 5B,C). Losartan significantly decreased M1 macrophages and the ratio of M1/M2 macrophages (CD11c/CD206, CCR7/CD163) in *ob*/*ob* mice (Figure 5B,C). IL-1β saw no significant changes in *ob*/*ob* mice treated with losartan (Figure 5A). HIF-1α protein was determined in the epididymal fat tissue homogenization in a Western blot (Figure 2), that is associated with increased angiogenesis markers, transforming growth factor beta (TGF-β), vascular endothelial growth factor (VEGF), and matrix metallopeptidase 9 (MMP-9) protein levels in the liver of *ob*/*ob* mice, suggesting enhanced function of HIF-1 in the liver (Figure 5D,E). Losartan was significantly reduced by angiogenesis markers, TGFβR2, VEGF, and MMP9 in *ob*/*ob* mice (Figure 5D,E). Collectively, these results suggested that losartan might play a certain role in regulation of liver inflammation and macrophage polarization in *ob*/*ob* mice.

### 2.6. Losartan Attenuates Lipid Accumulate and HIF-1α Protein Level in EWAT

To further study the regulation of LDs as dynamic lipid storage organelles that are found in EWAT on *ob*/*ob* mice, we assessed the adipocytes morphology, HIF-1α, and LD-associated protein levels (Figure 6). We have shown that HIF-1α protein levels and the size of lipid droplets were increased in *ob*/*ob* mice, and these observations were attenuated in EWAT of *ob*/*ob* mice treatment with losartan (Figure 6A,B). LD-associated proteins (PLIN1, PLIN2, CIDEA, and CIDEC) levels were significantly increased in *ob*/*ob* mice (Figure 6C,D). Losartan significantly reduced LD-associated proteins in *ob*/*ob* mice, but PLIN2 saw no significant change by western blot analysis (Figure 6D).

### 2.7. Losartan Attenuates Inflammation and Alters Macrophage Polarization in EWAT

We aimed to investigate whether losartan might be related to inflammatory responses in EWAT. We examined inflammatory gene expression and macrophage accumulation in EWAT of *ob*/*ob* mice. The mRNA expression of IL-1β, MCP1, TGFβ, TNFα, and IFNγ were increased in *ob*/*ob* mice, and inflammatory gene expression was significantly reduced in *ob*/*ob* mice treated with losartan (Figure 7A). Next, we showed that the effects of the macrophage polarization observed in EWAT of *ob*/*ob* mice treatment with losartan. M1 macrophage markers F4/80, CD11b, CD11c, and CCR7 were increased and M2 macrophage markers CD206 and CD163 were decreased in EWAT of *ob*/*ob* mice as compared to normal mice. We also observed that alternatively activated M2 markers (CD206, CD163) were upregulated and M1 markers (F4/80, CD11b, CD11c, CCR7) were downregulated in EWAT of *ob*/*ob* mice treated losartan (Figure 7B,C). Together, we demonstrated that losartan reduced inflammatory cytokines, M1 macrophage activation, and enhanced M2 macrophages activation in EWAT of *ob*/*ob* mice.

## 3. Discussion

In the present study, we evaluated the effects of losartan on NAFLD in an obese model; there was no significant improvement in plasma ALT and AST, which is in line with the previous observation of Hirata and coworkers in NAFLD patients [23]. However, losartan is able to decrease plasma TG, FFA, and the liver-to-body (L/B) weight ratio in *ob*/*ob* mice, which suggests that losartan might improve fat deposition in the liver. In addition, several other pathways are also likely to be involved in the beneficial effects of losartan, which are well known in NASH progression and are based on inflammation processes and TG accumulation. Losartan treatment strongly reduced steatosis score [21], as well as decreased oxidative stress and liver fibrosis in nonalcoholic steatohepatitis in rats [22]. However, the exact mechanism of losartan therapy for NAFLD is unclear, which is needed to elucidate further studies.

In liver steatosis, swelling of hepatocytes caused by lipid accumulation results in decreased hepatic sinusoidal perfusion and impaired hepatic microcirculation, thereby accelerating hepatic hypoxia [27]. As the homeostatic response to hypoxia increased to molecular genetic mechanisms, in which HIF-1 and HIF-2 have important roles, hypoxia induces liver lipid accumulation of metabolic adaptation [28]. Hepatic HIF-1α is activated in rodent models of diet-induced liver steatosis [28] and in human beings with NAFLD [29]. Early findings demonstrate that HIF-1 is implicated in lipid metabolism through LD accumulation [5], increase of fatty acid and lipid synthesis [5,30], and upregulation of fatty acid uptake [7], but the effect of LD-associated proteins is unclear. Inactivation of HIF-2α significantly suppressed the development of hepatic steatosis, indicating a novel role for HIF-2 in the regulation of hepatic lipid metabolism in vivo [31]. However, controversy remains regarding the role of HIF-2 as a pro-lipogenic factor [32], in as much as HIF-2 -deficient mice also exhibit hepatic steatosis, and the forced expression of HIF-1, but not HIF-2, in liver stimulates lipid accumulation in mice [33]. Therefore, we sought to elucidate the link between hepatic HIF-1α activation and LD-associated proteins. LD-associated proteins are essential for the maintenance of development of hepatic steatosis in mice [5,30,33]. Several proteins located at the surface of LDs (hypoxia-inducible protein 2, HIG2; Perilipin; PLIN2/adipose differentiation-related protein, ADRP; and Tip47) are essential for their membrane integrity, and HIG2 and ADRP are induced by hypoxia [5,30,33].

In addition, several reports showed that the CIDE family proteins are important regulators of various aspects of lipid metabolism, including control of lipid storage and LD size of adipocytes (by Cidea and Fsp27) [34,35]. Cidea deficient mice have lean phenotypes and are resistant to obesity [36], and there was up-regulation of mitochondrial activity and acquirement of brown adipose tissue-like properties in the white adipose tissue of CIDEC/FSP27 deficient mice [34,35,36]. Furthermore, liver-specific knocking down of Cidea in *ob*/*ob* mice resulted in less lipid accumulation [37], and alleviated hepatic steatosis SREBP-1c stimulated the transcription of Cidea by directly binding to the SRE identified in the Cidea gene promoter in hepatocytes [34,37]. Furuta et al. reported that the fatty acid synthase (FAS) gene is up-regulated by hypoxia via activation of Akt and SREBP-1 [38]. In the present study, our results agreed with previously published evidence, in which losartan decreased hepatosteatosis [21] and PLIN2 [39] in the liver of an obesity model.

Our current observations suggest that losartan plays a role in attenuating hepatic steatosis and hypoxia in the initiation of lipid accumulation. The protein levels of HIF-1α and LD-associated proteins were significantly decreased in liver and EWAT of *ob*/*ob* mice treated with losartan. Here, we delineate the potential mechanism of losartan that inhibits hepatic lipogenesis, and would attenuate hepatosteatosis in obesity-associated disease.

This study revealed that hepatic steatosis is accompanied not only by lipogenesis but also by impaired mitochondrial biogenesis. Previous studies have demonstrated that mitochondrial activity is globally repressed in more severe models of obesity, such as *ob*/*ob* and *db*/*db* mice, and Gao Q et al. conclude that PPARα and fatty acid oxidation (LCAD and MCAD) was increased in *ob*/*ob* mice [40]. These mice have lower arterial oxygenation suggestive of mild hypoxia and exposure to higher plasma cytokine levels, as well as HIF-1α being stabilized and promotion of mitochondrial complex IV dysfunction (decreased activity and stability) in age-dependent obesity [41,42]. Incomplete mitochondrial and lipid oxidation and/or respiratory chain dysfunction, when combined with limited antioxidant activity, increases hepatic oxidative stress and liver injury prior to the development of obesity [40,41,42].

We showed that protein level analysis confirmed downregulation of mitochondrial biogenesis and upregulation β-oxidation, suggesting mitochondrial dysfunction of livers in *ob*/*ob* mice [42,43]. As a result, the expression levels of key regulatory factors involved in mitochondrial metabolism and organelle biogenesis, namely, PGC-1α, TFAM, and NRF-2, have been reported to be reduced in NAFLD [43,44,45]. Losartan significantly increased mitochondrial biogenesis (SIRT1, PGC1α, UCP1, UCP2 protein levels, and PGC1α, NRF1, NRF2, and TFAM mRNA expression), and reduced β-oxidation was found in livers from *ob*/*ob* mice. Rodgers JT et al. confirmed the SIRT1 was down-regulated due to decreased NAD^+^ levels, which allowed the acetylation and activation of HIF-1a during hypoxia [45,46,47]. In addition, the SIRT1-HIF-1α interaction in hypoxic mouse tissues and observed in vivo showed that SIRT1 has negative effects on tumor growth and angiogenesis [44,45,46,47,48,49]. Accordingly, the liver might promote steatosis by enhancing HIF-1α and inhibiting SIRT1 signaling that stimulates hepatosteatosis in *ob*/*ob* mice, and these phenomena were neutralized by losartan.

Another mechanism impairing mitochondrial respiration involves ROS-mediated release of TNF in the liver [48,49]; Van den Bossche et al. showed that inflammatory M1 macrophage activation dampens mitochondrial function, thereby preventing the repolarization to an anti-inflammatory M2 phenotype [48]. HIFs have recently been identified as important regulators of immunity and inflammation. Wang et al. showed that macrophage activation in NASH involves a complex interplay between HIF-1α and autophagy, as these pathways promote pro-inflammatory overactivation in MCD diet-induced NASH [49,50]. In this study, we found that HIF-1α mediates MCP1, TNFα, and IFNγ production and that HIF-1α-mediated impairment of liver and EWAT increases IL-1β production [49,50,51], which induced M1 polarization [19,51] in mice contributing to obesity-induced NAFLD. We were able to detect expression of M1 macrophage (CD11b, CD11c, CCR7) protein levels, and the ratio of M1/M2 was increased in the liver and EWAT of *ob*/*ob* mice. In this study, losartan is involved in dampening obesity-induced proinflammatory gene expression, suppressing the ratio of M1/M2, suggesting that HIF-1α activation plays a prominent role in related macrophage polarization in *ob*/*ob* mice. The link between hypoxia and macrophage polarization in hepatosteatosis and EWAT suggests that losartan may interfere with the progression of obesity-associated disease.

This current study demonstrates the role of losartan in improving steatosis in *ob*/*ob* mice significantly: (1) losartan attenuated lipogenesis and LDs accumulation in hepatosteatosis and EWAT; (2) losartan improved mitochondrial biogenesis and β-oxidation in hepatosteatosis; (3) losartan decreased the inflammatory response and macrophage polarization in hepatosteatosis and EWAT. In conclusion, this study demonstrates that losartan attenuated hepatic LDs accumulation and enhanced mitochondrial function and M2 macrophage polarization, possibly through partial effects by modulating HIF-1α signaling, and is a major link between obesity and NAFLD. Thus, our study identifies that losartan could be a novel therapy for the treatment of NAFLD, which has become a worldwide health-threatening epidemic.

## 4. Materials and Methods

### 4.1. Animal Experiments

C57BL/6J (normal) mice and leptin-deficient (*ob*/*ob*) mice, obtained from the National Laboratory Animal Center (Taiwan), were housed for this study. All animal procedures were performed according to standard protocols and in compliance with standard recommendations for the proper care and use of laboratory animals. This study was approved by the Animal Care and Use Committee of Chang Gung University Institutional Animal Care and Use Committee (approved serial no. CGU14-044). All of the mice were group-housed (four to five mice per cage) in Macrolon cages in clean-conventional animal rooms in the American Association for Accreditation of Laboratory Animal Care (AAALAC)-accredited animal facility at Chang Gung University (12-h light-dark cycle, relative humidity 50–60%, and temperature ~21 °C) and had ad libitum access to food and water. The mice were then matched into three groups (*n* = 5/group) based on body weight: (1) normal mice, (2) *ob*/*ob* mice; (3) *ob*/*ob* mice+ losartan (100 mg/L dissolved in water for 30 days). After 30 days, animals were terminated by gradual fill CO_2_ asphyxiation, and a terminal blood sample was collected by cardiac puncture.

### 4.2. Sample Collection and Biochemical Analysis

Plasma was obtained by centrifugation and stored at −80 °C until biochemical analysis. Plasma alanine aminotransferase (ALT) and AST (Aspartate aminotransferase, GOT) were measured by Randox ATL and AST assay (RANDOX, Antrim, UK). Plasma triglycerides (TG) and free fatty acids (FFA) were measured by Randox triglycerides assay and NEFA Assay.

### 4.3. Histological, Immunohistochemistry, and Immunofluorescence Analysis of Adipose Tissue and Liver

Liver and EWAT were fixed overnight in 10% phosphate-buffered formalin, and were embedded in paraffin. Five-micrometer-thick sections were then stained with hematoxylin and eosin (HE) to assess lipid infiltration. For immunohistochemistry and immunofluorescence, liver and EWAT sections were stained with anti-PLIN1 (Ab3526, abcam, Cambridge, UK), PLIN2 (NB110-40877, Novus Biologicals, Centennial, CO, USA), CIDEA (NBP1-76950, Novus Biologicals, Centennial, CO, USA), CIDEC (Ab198204, abcam, Cambridge, UK), SREBP-1 (SC-366, Santa Cruz Biotechnology, Dallas, TX, USA), CD36 (SC-70644, Santa Cruz Biotechnology, Dallas, TX, USA), SIRT1 (ab110304, abcam, Cambridge, UK), PGC1α (ab54481, abcam, Cambridge, UK), UCP1 (ab10983, abcam, Cambridge, UK), HIF-1α (ab179483, abcam, Cambridge, UK), TGFβR2 (Santa Cruz Biotechnology, Dallas, TX, USA), VEGF (ab69479, abcam, Cambridge, UK), MMP9 (AB19016, Millipore, Burlington, MA, USA), F4/80 (Ab6640, abcam, Cambridge, UK), CD11b (ab133357, abcam, Cambridge, UK), CD11c (ab52632, ab11029, abcam, Cambridge, UK), CCR7 (abcam, Cambridge, UK), CD163 (ab182422, abcam, Cambridge, UK), CD206 (ab64693, abcam, Cambridge, UK), DAPI (62248, Thermo Fisher Scientific, Waltham, MA, USA) and MitoTracker Red CMXRos (M7512, Thermo Fisher Scientific, Waltham, MA, USA). For immunohistochemistry analysis, sections were also stained with secondary antibodies conjugated with HRP-conjugated anti-rabbit (G-21234, Millipore, Burlington, MA, USA), anti-mouse (G-21040, Millipore, Burlington, MA, USA), anti-rat (31470, Genetex, Irvine, CA, USA) or anti-goat (31402, abcam, CA, UK), followed by incubation in DAB peroxidase solution (Millipore, Burlington, MA, USA) and subsequent counterstaining with hematoxylin (Sigma, St. Louis, MO, USA). The images shown here were obtained using an Olympus IX71 microscope. For immunofluorescence analysis, sections were incubated with the secondary antibody, mouse secondary antibody Alexa Fluor 488 (A-11029, Thermo Fisher Scientific, Waltham, MA, USA), rabbit Secondary Antibody, Alexa Fluor 488 (A11034, Thermo Fisher Scientific, Waltham, MA, USA), rabbit Secondary Antibody, Alexa Fluor Plus 647 (A32795, Thermo Fisher Scientific, Waltham, MA, USA), goat Alexa Fluor 488, Alexa Fluor 633, anti-mouse or anti-rabbit (Thermo Fisher Scientific, Waltham, MA, USA) and DAPI (62248, Thermo Fisher Scientific, Waltham, MA, USA) for nuclear staining. Positive staining for CD11c, CCR7, CD163, CD206 were quantified using ImageJ software (1.45, NIH).

### 4.4. Oil Red O Staining

Lipid droplets were visualized by Oil Red O staining (ORO; Sigma-Aldrich; Merck KGaA, Darmstadt, Germany). Fresh liver tissues were embedded carefully in an optimal cutting temperature compound (OCT) in a plastic mold, followed by freezing at −80 °C. Liver tissue sections (10 μm thick) were stained with Oil Red O working solution (*w*/*v*, 60% isopropyl alcohol and 40% water) for 15 min, that was later rinsed with 50% isopropanol and counterstained with hematoxylin for the nucleus.

### 4.5. Auto-Fluorescence Detection of Lipofuscin

The liver sections were deparaffinized, hydrated and mounted into 40% glycerol/TBS mounting medium. Lipofuscin auto-fluorescence was then evidenced by excitation at 450–490 nm, using a dichromatic mirror at 510 nm and a long-pass filter at 515 nm [52].

### 4.6. RNA Isolation and Quantitative Real-Time PCR

The liver and EWAT samples were obtained from all mice. Total RNAs of liver and EWAT tissues were extracted using TRIzol Reagent (Life Technologies, Waltham, MA, USA) and a RNeasy Kit (QIAGEN, Germantown, MD, USA). The complementary DNA (cDNA) was obtained using a High-Capacity cDNA Reverse Transcription Kit (Thermo Fisher Scientific, Waltham, MA, USA). The cDNA samples were amplified by a quantitative reverse-transcription polymerase chain reaction using Fast SYBR Green (Roche, Basel, Switzerland). Relative expression levels were determined by normalizing each Ct value to either gene expression for mice samples using the △△Ct method. The primer sequences used in this study are shown in Table 1.

### 4.7. Protein Isolation and Western Blot Analysis

Western blotting was performed using 50–100 μg of nuclear fraction, cytosolic fraction or whole extract of liver and EWAT. Tissues were homogenized in lysis buffer (100 mM Tris-HCl, pH 7.6, 2mM EDTA, 2mM EGTA, 150 mM NaCl, 1% Triton X-100) containing proteinase inhibitors and phosphatase inhibitors (78442, Thermo Fisher Scientific, Waltham, MA, USA). The specific antibodies against the respective antigens were as follows: PLIN1 (ab3526, abcam, Cambridge, UK), PLIN2 (NB110-40877, Novus Biologicals, Centennial, CO, USA USA), CIDEA (NBP1-76950, Novus Biologicals, Centennial, CO, USA), CIDEC (ab198204, abcam, Cambridge, UK), SREBP-1 (SC-366, Santa Cruz Biotechnology, Dallas, TX, USA), CD36 (SC-70644, Santa Cruz Biotechnology, Dallas, TX, USA), SIRT1 (ab110304, abcam, Cambridge, UK), PGC1α (ab54481, abcam, Cambridge, UK), UCP1 (ab10983, abcam, Cambridge, UK), UCP2 (ab203244, ab97931, abcam, Cambridge, UK) HIF1α (ab179483, abcam, Cambridge, UK), TGFβR2 (SC-17791, Santa Cruz Biotechnology, Dallas, TX, USA), VEGF (ab69479, abcam, Cambridge, UK), CD11b (ab133357, abcam, Cambridge, UK), CD11c (ab52632, ab11029, abcam, Cambridge, UK), CCR7 (ab32527, abcam, Cambridge, UK), CD163 (ab182422, abcam, Cambridge, UK), CD206 (ab64693, abcam, Cambridge, UK), β-actin (MAB1501, Millipore, Burlington, MA, USA), and Histone (SC-56616, Santa Cruz Biotechnology, Dallas, TX, USA). The protein expression was detected using an enhanced chemiluminescence kit (Millipore, Burlington, MA, USA), and quantified using ImageQuant 5.2 software. Each experiment was repeated with a minimum of three independently prepared protein samples.

### 4.8. Statistical Analysis

All data are expressed as means ± SEM. The significance of differences between values was determined using the two-tailed unpaired *t* test. *p* values less than 0.05 were considered significant.

## Figures and Tables

**Figure 1 ijms-22-07841-f001:**
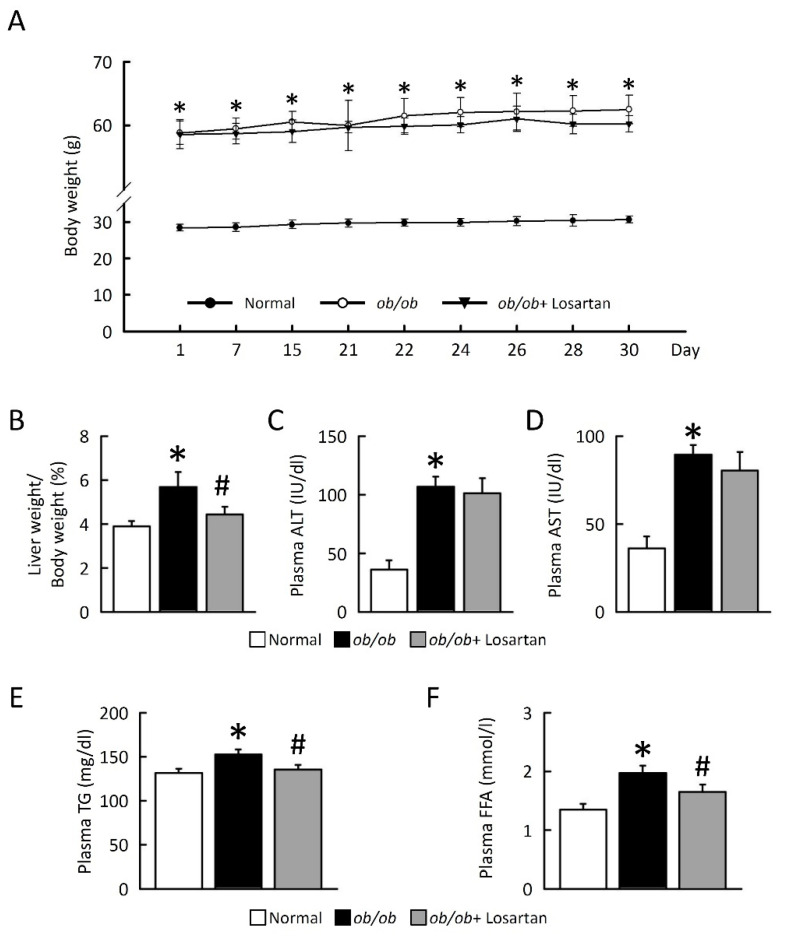
Losartan attenuated NAFLD development in *ob*/*ob* mice. (**A**) Body weight (g). (**B**) Liver weight/body weight (%). (**C**) Plasma ALT (IU/dL). (**D**) Plasma AST (IU/dL). (**E**) Plasma TG (mg/dL). (**F**) Plasma FFA (mmol/L). For each animal group, *n* = 5. All values represent the mean ± SEM. Data were analyzed by Student’s *t* test. * *p* ≤ 0.05; normal vs. *ob*/*ob*. # *p* ≤ 0.05; *ob*/*ob* vs. *ob*/*ob* + Losartan. ALT, alanine transaminase; AST, aspartate aminotransferase; TG, triglyceride; FFA, free fatty acid.

**Figure 2 ijms-22-07841-f002:**
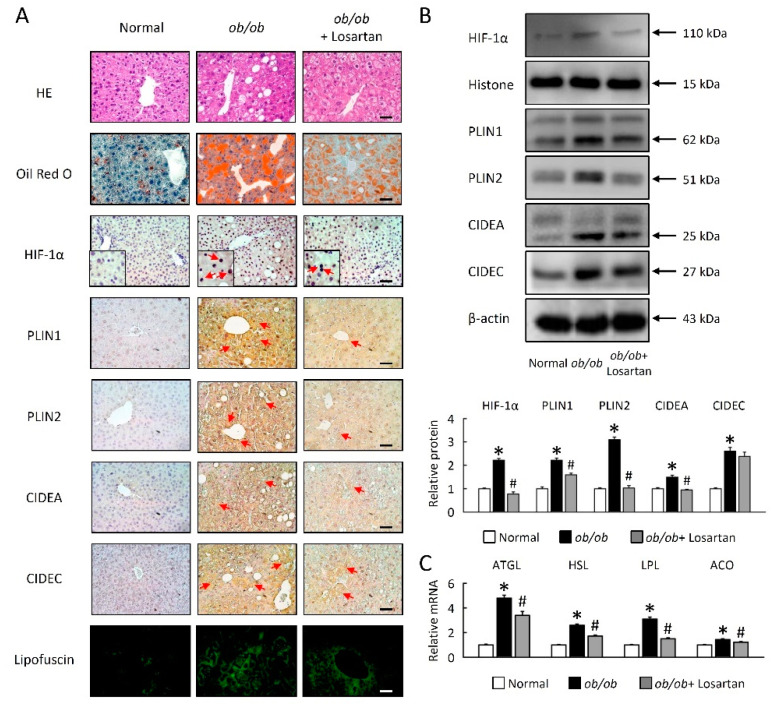
Losartan decreased HIF-1α and LD-associated proteins. (**A**) Representative HE, Oil Red O, HIF-1α, PLIN1, PLIN2, CIDEA and CIDEC staining of liver from normal mice and losartan-treated *ob*/*ob* mice. Green pseudo-color represents visualization of lipofuscin’s autofluorescence at 450–490 nm. Red arrow highlights the positive staining. Scale bar: 100 μm. Quantification of (**B**) HIF-1α, PLIN1, PLIN2, CIDEA, and CIDEC protein levels by Western blot of liver after losartan treatment. Below graphs indicate quantification relative to Histone or β-actin. (**C**) Quantification of ATGL, HSL, LPL, and ACO by qRT-PCR. qRT-PCR indicates quantification relative to GAPDH. For each animal group, *n* = 5. All values represent the mean ± SEM. Data were analyzed by Student’s *t* test. * *p* ≤ 0.05; normal vs. *ob*/*ob*. # *p* ≤ 0.05; *ob*/*ob* vs. *ob*/*ob* + Losartan. HIF-1α, hypoxia-inducible factor-1α; LDs, lipid droplets; HE, hematoxylin and eosin; PLIN, perilipin; CIDE, cell-death-inducing DNA-fragmentation-factor (DFF45)-like effector; ATGL, adipose triglyceride lipase; HSL, hormone-sensitive lipase; LPL, lipoprotein lipase; ACO, acyl-CoA oxidase.

**Figure 3 ijms-22-07841-f003:**
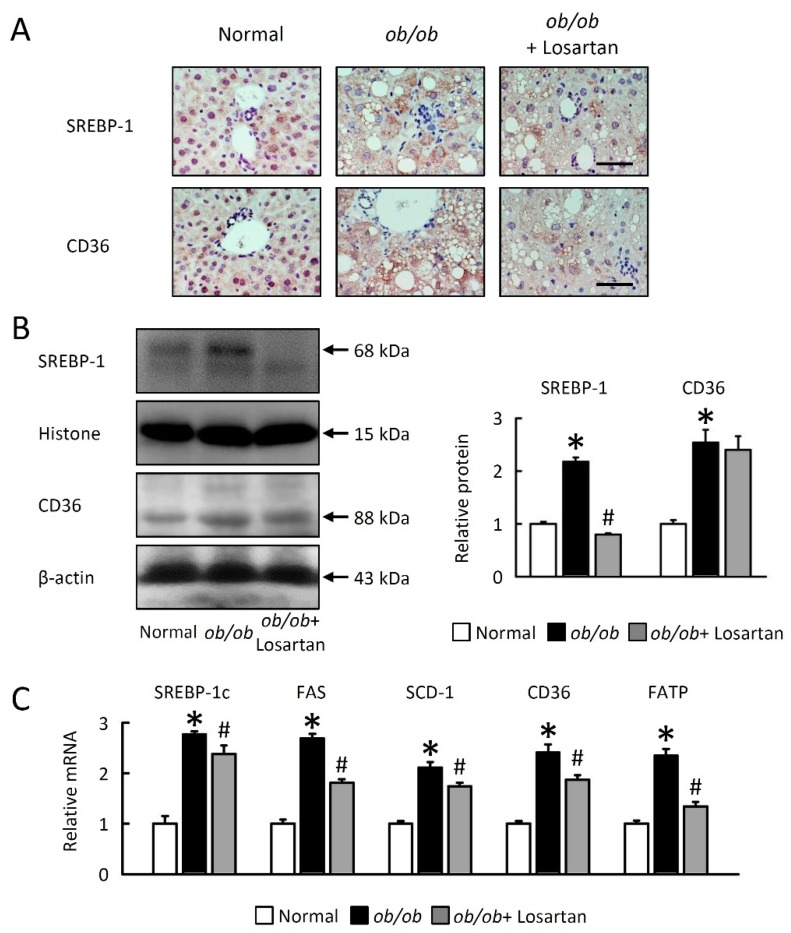
Losartan attenuates lipogenesis and improves lipolysis in liver. (**A**) Representative SREBP-1 and CD36 staining of liver from-treated *ob*/*ob* mice and normal mice. Scale bar: 100 μm. (**B**) Quantification of SREBP-1 and CD36 protein levels by Western blot of liver after losartan treatment. Right-hand graphs indicate quantification relative to Histone (for SREBP-1) and β-actin (for CD36). (**C**) Quantification of SREBP-1c, FAS and SCD-1, CD36, and FATP by qRT-PCR. qRT-PCR indicate quantification relative to GAPDH. For each animal group, *n* = 5. All values represent the mean ± SEM. Data were analyzed by Student’s t test. * *p* ≤ 0.05; normal vs. *ob*/*ob*. # *p* ≤ 0.05; *ob*/*ob* vs. *ob*/*ob* + Losartan. SREBP-1, sterol regulatory element binding protein 1; CD36, cluster of differentiation 36; FAS, fatty acid synthase; SCD-1, stearoyl-CoA desaturase-1; FATP, fatty acid transport protein.

**Figure 4 ijms-22-07841-f004:**
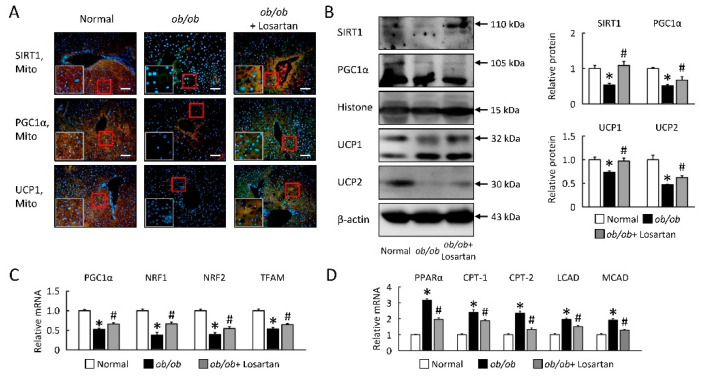
Losartan enhanced mitochondrial biogenesis and function in *ob*/*ob* mice. (**A**) The staining for SIRT1, PGC1α, and UCP1 are green, while the staining for mitochondria (Mito Tracker) is red. Staining for DAPI is blue. Magnification of tissue samples is 20× (red box) and for inset, magnification is 40× (white box). Scale bar: 100 μm. (**B**) Quantification of SIRT1, PGC1α, UCP1, and UCP2 protein levels by Western blot of liver after losartan treatment. Right graphs indicate quantification relative to Histone (for SIRT1 and PGC1α) and β-actin (for UCP1 and UCP2). Quantification of (**C**) PGC1α, NRF1, NRF2, TFAM, (**D**) PPARα, CPT-1, CPT-2, LCAD, and MCAD by qRT-PCR. qRT-PCR indicate quantification relative to GAPDH. For each animal group, *n* = 5. All values represent the mean ± SEM. Data were analyzed by Student’s *t* test. * *p* ≤ 0.05; normal vs. *ob*/*ob*. # *p* ≤ 0.05; *ob*/*ob* vs. *ob*/*ob* + Losartan. SIRT1, sirtuin-1; PGC1α, peroxisome proliferator-activated receptor gamma coactivator 1-alpha; PPAR, peroxisome proliferator-activated receptor; UCP1, uncoupling protein 1; NRF, nuclear respiratory factor; TFAM, mitochondrial transcription factor A; CPT, carnitine palmitoyltransferase; LCAD, long-chain acyl-CoA dehydrogenase; MCAD, medium-chain acyl-CoA dehydrogenase.

**Figure 5 ijms-22-07841-f005:**
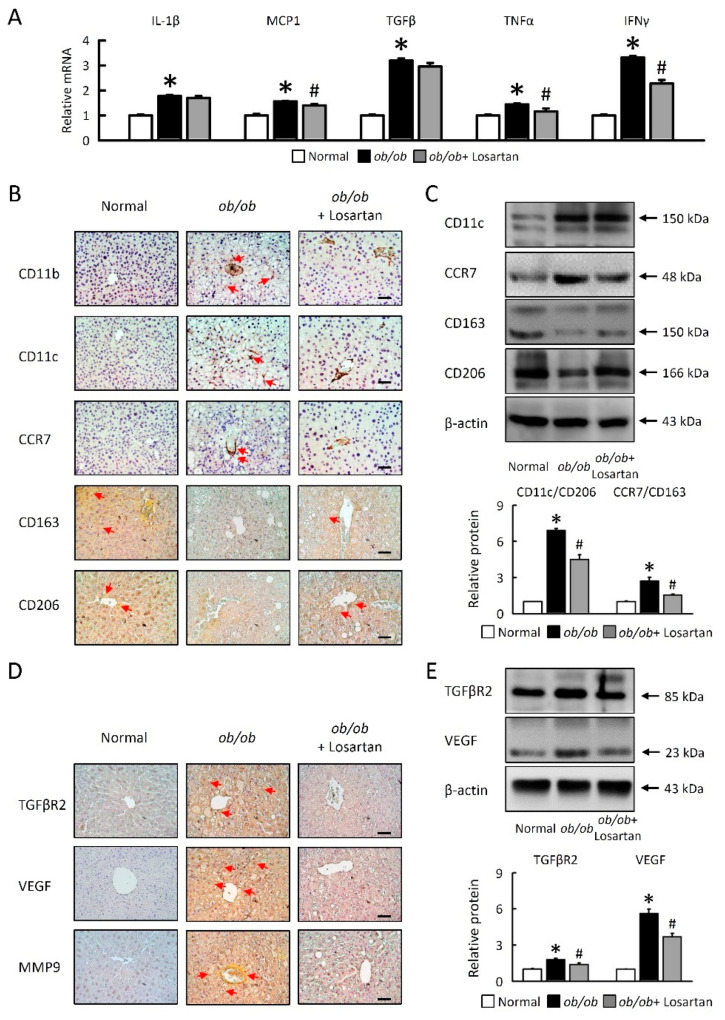
Losartan related macrophage polarization and angiogenesis in liver. (**A**) Quantification of IL-1β, MCP-1, TGFβ, TNFα, and IFNγ by qRT-PCR. qRT-PCR indicates quantification relative to GAPDH. (**B**) Representative CD11b, CD11c, CCR7, CD163 and CD206 staining of liver. Red arrow highlights the positive staining. Scale bar: 100 μm. (**C**) Quantification of CD11c, CCR7, CD163, and CD206 protein levels by Western blot of liver. Below graphs indicate quantification relative to β-actin. Ratio of CD11c at CD206 and CCR7 at CD163 in liver. (**D**) Representative TGFβR2, VEGF, and MMP9 staining of liver. Red arrow highlights the positive staining. Scale bar: 100 μm. (**E**) Quantification of TGFβR2 and VEGF protein levels by Western blot of liver. Below graphs indicate quantification relative to β-actin. For each animal group, *n* = 5. All values represent the mean ± SEM. Data were analyzed by Student’s *t* test. * *p* ≤ 0.05; normal vs. *ob*/*ob*. # *p* ≤ 0.05; *ob*/*ob* vs. *ob*/*ob* + Losartan. IL-1β, interleukin-1β; MCP-1, monocyte chemoattractant protein-1; TGFβ, transforming growth factor beta; TNFα, tumor necrosis factor α; IFNγ, interferon gamma; TGFβR2, transforming growth factor beta receptor 2; VEGF, targets vascular endothelial growth factor; MMP9, matrix metallopeptidase 9.

**Figure 6 ijms-22-07841-f006:**
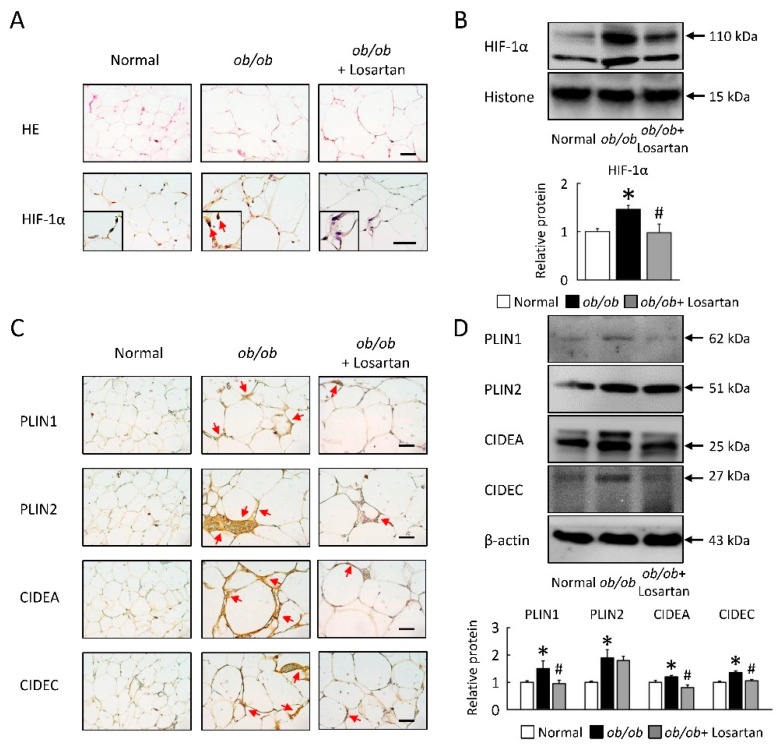
Losartan attenuated HIF-1α expression and LDs formation in EWAT. (**A**) Representative HIF-1α staining of EWAT from losartan-treated *ob*/*ob* mice and normal mice. Red arrow highlights the positive staining. Scale bar: 100 μm. Quantification of (**B**) HIF-1α protein level by Western blot. Below graphs indicate quantification relative to Histone. (**C**) Representative HE, PLIN1, PLIN2, CIDEA and CIDEC staining of EWAT from losartan-treated *ob*/*ob* mice and normal mice. Scale bar: 100 μm. Quantification of (**D**) PLIN1, PLIN2, CIDEA, and CIDEC protein levels by Western blot of EWAT after losartan treatment. Below graphs indicate quantification relative to β-actin. For each animal group, *n* = 5. All values represent the mean ± SEM. Data were analyzed by Student’s *t* test. * *p* ≤ 0.05; normal vs. *ob*/*ob*. # *p* ≤ 0.05; *ob*/*ob* vs. *ob*/*ob* + Losartan. HIF-1α, hypoxia-inducible factor-1α; LDs, lipid droplets; EWAT, epididymis white adipose tissue; HE, hematoxylin and eosin; PLIN, perilipin; CIDE, cell-death-inducing DNA-fragmentation-factor (DFF45)-like effector.

**Figure 7 ijms-22-07841-f007:**
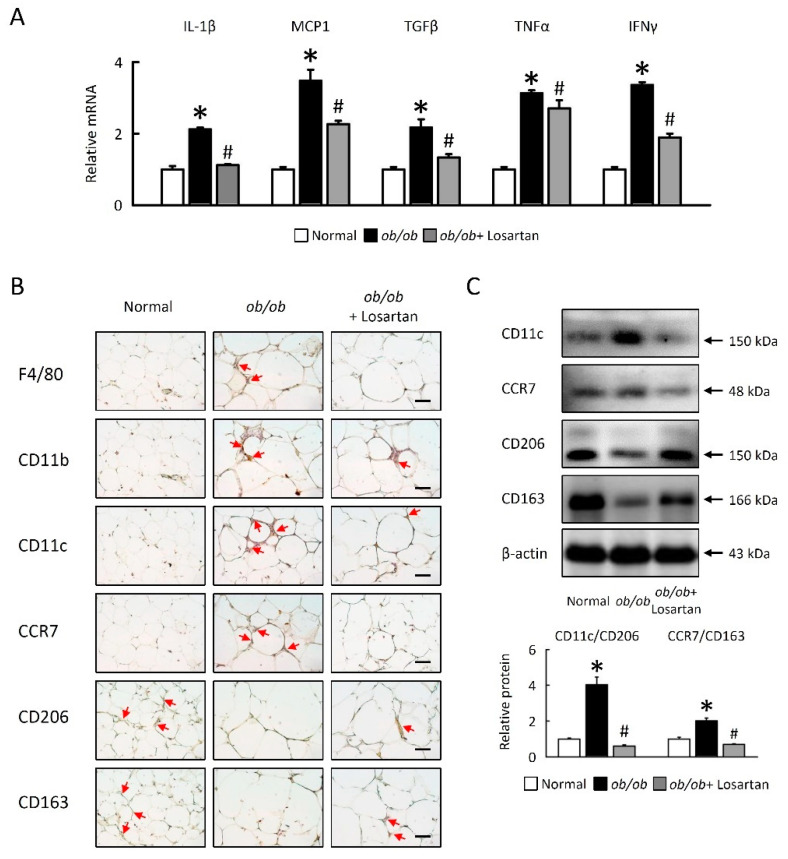
Losartan altered macrophage polarization in EWAT. (**A**) Quantification of IL-1β, MCP1, TGFβ, TNFα, and IFNγ by qRT-PCR. qRT-PCR indicates quantification relative to GAPDH. (**B**) Representative F4/80, CD11b, CD11c, CCR7, CD206 and CD163 staining of EWAT. Red arrow highlights the positive staining. Scale bar: 100 μm. (**C**) Quantification of CD11c, CCR7, CD206, and CD163 protein levels by Western blot of EWAT. Below graphs indicate quantification relative to β-actin. The ratio of CD11c at CD206 and CCR7 at CD163 in the EWAT. For each animal group, *n* = 5. All values represent the mean ± SEM. Data were analyzed by Student’s *t* test. * *p* ≤ 0.05; normal vs. *ob*/*ob*. # *p* ≤ 0.05; *ob*/*ob* vs. *ob*/*ob* + Losartan. EWAT, epididymal white adipose tissue; IL-1β, interleukin-1β; MCP1, monocyte chemoattractant protein-1 (CCL2); TGFβ, transforming growth factor beta; TNFα, tumor necrosis factor α; IFNγ, Interferon gamma.

**Table 1 ijms-22-07841-t001:** Oligonucleotide sequences for t-qPCR.

Gene	Forward	Reverse
*ATGL*	5′ aacaccagcatccagttcaa 3′	5′ ggttcagtaggccattcctc 3′
*HSL*	5′ agacaccagccaacggatac 3′	5′ catcaccctcgaagaagagca 3′
*LPL*	5′ actcatctccgccatgcc 3′	5′ ccagctttctcctagcaagg 3′
*ACO*	5′ atgaatcccgatctgcgcaaggagc 3′	5′ aaaggcatgtaacccgtagcactcc 3′
*SREBP-1c*	5′ actgtcttggttgttgatgagctggagcat 3′	5′ atcggcgcggaagctgtcggggtagcgtc 3′
*FAS*	5′ tgtcattggcctcctcaaaaagggcgtcca 3′	5′ tcaccactgtgggctctgcagagaagcgag 3′
*SCD-1*	5′ ccggagaccccttagatcga 3′	5′ tagcctgtaaaagatttctgcaaacc 3′
*FATP*	5′ gcttcaacagccgtatcctc 3′	5′ tcttcttgttggtggcactg 3′
*CD36*	5′ gcaaaacgactgcaggtcaac 3′	5′ tggtcccagtctcatttagcca 3′
*CPT-1*	5′ ggacagagactgtgcgttcct 3′	5′ gcgatatccaacagtgcttga 3′
*CPT-2*	5′ caaggccctggctgatgatgtg 3′	5′ agtctctgtccgcccctctcg 3′
*LCAD*	5′ tcaacagcagttacttgg 3′	5′ gacaatatctgagtggag 3′
*MCAD*	5′ ggggaggatgacggagcagc 3′	5′ cgggtactttaggatctggg 3′
*PGC1α*	5′ gactcagtgtcaccaccgaaa-3′	5′ tgaacgagagcgcatcctt 3′
*TFAM*	5′ ggaatgtggagcgtgctaaaa 3′	5′-tgctggaaaaacacttcggaata 3′
*UCP1*	5′ cctgcctctctcggaaacaa 3′	5′-tgtaggctgcccaatgaaca 3′
*UCP2*	5′ gcctctggaaagggacttctc 3′	5′ accagctcagcacagttgaca 3′
*NRF1*	5′ cgcagcacctttggagaa 3′	5′-cccgacctgtggaatacttg-3′
*NRF2*	5′ atggatttgattgacatcctt 3′	5′ catgtttttctttgtatctgg 3′
*IL-1β*	5′ aacctgctggtgtgtgacgttc 3′	5′ cagcacgaggcttttttgttgt 3′
*MCP1*	5′ aggtccctgtcatgcttctg 3′	5′ tctggacccattccttcttg 3′
*TGFβ*	5′ tatagcaacaattcctggcg 3′	5′ tgctgtcacaggagcagtg 3′
*TNFα*	5′ ttgacctcagcgctgagttg 3′	5′ cctgtagcccacgtcgtagc 3′
*IFNγ*	5′ cctcaaacttggcaatactc 3′	5′ agcaacaacataagcgtcat 3′
*GAPDH*	5′ tcaccaccatggagaaggc 3′	5′ gctaagcagttggtggtgca 3′

ATGL, adipose triglyceride lipase; HSL, hormone-sensitive lipase; LPL, lipoprotein lipase; ACO, acyl-CoA oxidase; SREBP-1c, sterol regulatory element-binding protein 1c; FAS, fatty acid synthase, SCD-1, stearoyl-CoA desaturase-1, FATP, fatty acid transport protein; CD36, cluster of differentiation 36; CPT-1, carnitine palmitoyltransferase 1; LCAD, long-chain acyl-CoA dehydrogenase; MCAD, medium-chain acyl-CoA dehydrogenase (ACADM); PGC1α, peroxisome proliferator-activated receptor gamma coactivator 1-alpha; TFAM, mitochondrial transcription factor A; UCP1, uncoupling protein 1; NRF1, nuclear respiratory factor 1; IL-1β, interleukin-1β; MCP1, monocyte chemoattractant protein-1 (CCL2); TGFβ, transforming growth factor beta; TNFα, tumor necrosis factor α; IFNγ, interferon gamma; GAPDH, glyceraldehyde-3-phosphate dehydrogenase.

## Data Availability

The data generated for this study are available from the corresponding author on reasonable request.

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
