# Peer review of "Losartan Prevents Hepatic Steatosis and Macrophage Polarization by Inhibiting HIF-1α in a Murine Model of NAFLD"

_ijms, 2021, doi:10.3390/ijms22157841_

Round 1
Reviewer 1 Report
In this study the effect of Losartan on hepatic steatosis and macrophage polarization in a murine model of NAFLD (ob/ob mice) is evaluated. The authors report that losartan attenuates the lipogenesis and lipid droplets (LDs) accumulation in hepatosteatosis and epididymal white adipose tissue (EWAT) in ob/ob mice Also, losartan improved mitochondrial biogenesis and β-oxidation in hepatosteatosis and decreased the inflammatory response and macrophage polarization in hepatosteatosis and EWAT. It is concluded that losartan attenuates hepatic LDs accumulation and enhances mitochondrial function and M2 macrophage polarization by inhibiting HIF-1α signaling and losartan could be a novel therapy for the treatment of NAFLD.
Unfortunately, in the ACROBAR copy of this paper that I received for reweaving the Figs, 3 and 4 were lacking. However, numerous weaknesses are present in the remaining text. The histology picture of Fig. 1A is too small and the alleged differences are not assessable while in other figures they are hardly visible. The Western blot show only single examples of “normal ob/ob and ob/ob + Loxartan” and the figures are too obscure so that the differences are not evaluable. The same problem regards the fig.s 5 and 6. The Discussion contains some unclear sentences: i.e. “We demonstrate M1 and M2 macrophages markers by immunofluorescence (IF) and immunohistochemistry (IHC) staining, that similar Zhang Q et al. [49] and Sarig U et al. studies [50]”. Furthermore, the curative effect of losartan in NAFLD was object of previous studies (Rosselli et al. Atherosclerosis. 2009, 206:119-26; Hirata et al. Int J Endocrinol. 2013;2013:587140; Kaji et al. Am J Physiol Gastrointest Liver Physiol. 2011;300:1094-104) that should be quoted and commented.
Author Response
Response to Reviewer 1
- Unfortunately, in the ACROBAR copy of this paper that I received for reweaving the Figs, 3 and 4 were lacking.
Response: All the figures have been rearranged and placed at the end of revised manuscript. (Page 19-25) - However, numerous weaknesses are present in the remaining text.
Response: Thank you Reviewer for pointing our weaknesses out. We have invited our foreign colleague to correct our weaknesses in the text and carefully revised current manuscript. - The histology picture of Fig. 1A is too small and the alleged differences are not assessable while in other figures they are hardly visible.
Response: All the figures have been rearranged and placed at the end of revised manuscript. (Page 19-25) - The Western blot show only single examples of “normal, ob/ob and ob/ob + Loxartan” and the figures are too obscure so that the differences are not evaluable.
Response: All the figures have been rearranged and placed at the end of revised manuscript. (Page 19-25). In addition, the Western blot images also mark up with molecular weight. - The same problem regards the s 5 and 6.
Response: All the figures have been rearranged and placed at the end of revised manuscript. (Page 19-25). In addition, the Western blot images also mark up with molecular weight. - The Discussion contains some unclear sentences: i.e. “We demonstrate M1 and M2 macrophages markers by immunofluorescence (IF) and immunohistochemistry (IHC) staining, that similar Zhang Q et al. [49] and Sarig U et al. studies [50]”.
Response: To avoid unnecessary or unclear meaning, we have deleted this paragraph. In addition, we have also made some corrections on the Discussion content of revised manuscript. (Page 12, line 379-385) - Furthermore, the curative effect of losartan in NAFLD was object of previous studies (Rosselli et al. Atherosclerosis. 2009, 206:119-26; Hirata et al. Int J Endocrinol. 2013;2013:587140; Kaji et al. Am J Physiol Gastrointest Liver Physiol. 2011;300:1094-104) that should be quoted and commented.
Response: In light of losartan specific antihypertensive effect, and its impact on liver fibrosis. As suggested by Reviewer, the evidence has added in references section (R21,22,23). We also addressed some points in the Introduction (Page 2, line 77-78) and Discussion (Page 10, line 275-283) and bring consistent results with our data in NAFLD mice model.
Please see the attached file is our revised manuscript.

Reviewer 2 Report
In the manuscript entitled “Losartan prevents hepatic steatosis and macrophage polarization by inhibiting HIF-1α in a murine model of NAFLD” authors have demonstrated the application of angiotensin II receptor antagonist, Losartan in the treatment of obesity-related steatosis and non-alcoholic fatty liver disease by enhancing hepatic mitochondrial biogenesis/function and M2 macrophage polarization by inhibiting HIF-1α and lipogenesis pathway.
Overall, this manuscript will be of interest to the people working in NAFLD. But certain points that need to be addressed to strengthen the manuscript are:
- Authors have shown that losartan decreases fat accumulation in liver, so authors should show liver weight/body weight ratio to further confirm the decrease in fat accumulation.
- Authors should also include serum lipid profile and ALT, AST levels.
- HE and Oil Red O staining images do not corroborate with each other. In the Oil Red O staining image more lipid droplets can be seen as compared to HE staining images. For Oil Red O staining lot of lipid droplets is seen in normal tissue also.
- Authors should also confirm the results of qRT-PCR by western blotting to ensure the changes they are seeing at mRNA are also seen at protein level.
- In qRT-PCR authors have not mentioned to which internal control they have normalized.
- Authors could comment if they have seen any effect of Losartan on HIF-2α, as it influences the steatohepatitis by altering lipid metabolism.
- Discussion part is not well written, it is confusing, authors could re-write this part to make it clearer.
- Authors have mentioned that losartan inhibited HIF-1α signaling, it would not be a correct statement, as in this manuscript authors have demonstrated effect on the proteins that are markers for lipolysis or lipogenesis or β-oxidation.
Minors:
- Authors could add molecular weight markers for the western blots
- Alignment of graph is disturbed for fig. 4E
- Typo error in line 317, “ration” should be replaced with ratio
- Word is missing in between line 370 and 371
- Remove “then” from line 377
- Authors could add catalog numbers for antibodies they have used
Author Response
Response to Reviewer 2
Major:
- Authors have shown that losartan decreases fat accumulation in liver, so authors should show liver weight/body weight ratio to further confirm the decrease in fat accumulation.
Response: Thank you for Reviewer’s suggestion liver weight/body weight ratio (Figure 1) has been rearranged and placed at the end of revised manuscript. (Page 19). - Authors should also include serum lipid profile and ALT, AST levels.
Response: In light of Reviewer’s suggestion, losartan can be beneficial in treating NAFLD and its consequences, we bring serum lipid profile and ALT, AST data (Figure 1) in the revised manuscript (Page 19). - HE and Oil Red O staining images do not corroborate with each other. In the Oil Red O staining image more lipid droplets can be seen as compared to HE staining images. For Oil Red O staining lot of lipid droplets is seen in normal tissue also.
Response: Oil Red O staining image (Figure 2) has been rearranged and placed at the end of revised manuscript (Page 20).. - Authors should also confirm the results of qRT-PCR by western blotting to ensure the changes they are seeing at mRNA are also seen at protein level.
Response: We do analyze several relevant and important indicators at protein and mRNA levels at our experimental design ( Fig. 3,4,5). The are indeed a few indicators that have not been assayed for protein scores. We hope Reviewer can understand that we cannot provide a complete analysis in such a short response time. - In qRT-PCR authors have not mentioned to which internal control they have normalized.
Response: GAPDH was used as internal control genes in qRT-PCR assay. We have addressed this issue in figure legend of entire revised manuscript. - Authors could comment if they have seen any effect of Losartan on HIF-2α, as it influences the steatohepatitis by altering lipid metabolism.
Response: As Reviewer’s suggestion, we have made some comments about the role of HIF-2α in steatohepatitis. (Page 10, line 290-292; line 297-302) - Discussion part is not well written, it is confusing, authors could re-write this part to make it clearer.
Response: Thank you Reviewer for pointing our weaknesses out. We have invited our foreign colleague to correct our weaknesses in the text and to avoid unnecessary or unclear meaning, we have deleted some paragraphs. In addition, we have also made some corrections in the Discussion content of revised manuscript. - Authors have mentioned that losartan inhibited HIF-1α signaling, it would not be a correct statement, as in this manuscript authors have demonstrated effect on the proteins that are markers for lipolysis or lipogenesis or β-oxidation.
Response: To avoid unnecessary or unclear meaning, we have made a correction statement “losartan attenuated hepatic LDs accumulation and enhanced mitochondrial function and M2 macrophage polarization possibly through partial effects by modulating HIF-1α signaling”. (Page 12, line 394-396)
Minors:
- Authors could add molecular weight markers for the western blots
Response: All the Western blot images mark up with molecular weight. Figures have been rearranged and placed at the end of revised manuscript. (Page 19-25). - Alignment of graph is disturbed for fig. 4E
Response: Figure 4E has been rearranged into 5E and placed at the end of revised manuscript. (Page 23). - Typo error in line 317, “ration” should be replaced with ratio
Response: The error has been corrected (Page 12, line 377). - Word is missing in between line 370 and 371
Response: The error has been corrected. - Remove “then” from line 377
Response: The error has been removed (Page 13, line 453). - Authors could add catalog numbers for antibodies they have used
Response: All the antibodies catalog numbers have been added in Materials and Methods section in revised manuscript (Page 13, 14, 15).
Please see attached file is our revised manuscript.

Round 2
Reviewer 1 Report
The authors have satisfactorily met the reviewers comments.
Reviewer 2 Report
Authors should check for grammar and spelling error especially in Introduction and Discussion part.
Authors can reframe the discussion part as it is still confusing.